# Anaemia and associated factors in people living with HIV on ART in Southern Province of Zambia

Martin Chakulya[1,2]*, Lweendo Muchaili[1,2], Hanzooma Hatwiko[1,2], Matenge Mutalange[1], Memory Ngosa[1], Bislom C. Mweene[1,2], Geofrey Mupeta[1], David Chisompola[1], Marshall C. Mubanga[1], Lukundo Siame[1,2,3], Joreen P. Povia[4], Benson M. Hamooya[4,5], Sepiso K. Masenga[1,2]*

1 Department of Pathology and Microbiology, Mulungushi University, School of Medicine and Health Sciences, Livingstone, Zambia, 2 Department of Cardiovascular Science and Metabolic Diseases, Livingstone Center for Prevention and Translational Science, Livingstone, Zambia, 3 Department of Internal Medicine, Livingstone University Teaching Hospital, Livingstone, Zambia, 4 Department of Health Economics, Livingstone Center for Prevention and Translational Science, Livingstone, Zambia, 5 Department of Public Health, Mulungushi University, School of Medicine and Health Sciences, Livingstone, Zambia

* chakulyamartin1@gmail.com (MC); sepisomasenga@lcpts.org (SKM)

## Abstract

### Background

Anaemia remains a significant comorbidity among people living with HIV (PLHIV) on antiretroviral therapy (ART) in Sub-Saharan Africa (SSA), exacerbated by biological, sociocultural, and clinical factors. Despite known sex disparities, limited data exist on sex-specific determinants of anaemia in Zambia. This study evaluates anaemia prevalence, sex differences, and associated factors among PLHIV in Southern Province, Zambia.

### Methods

A retrospective cohort study analyzed 2,840 PLHIV aged ≥15 years from 12 districts in Southern Zambia. Data were abstracted from medical records (November–December 2024). Anaemia was defined as haemoglobin <13 g/dL (men) and <12 g/dL (non-pregnant women). Multivariable logistic regression identified factors associated with anaemia.

### Results

Anaemia prevalence was 15.7% (446/2,840), with marked sex disparity: females exhibited 21.8% prevalence (380/1,741) versus 6.0% in males (66/1,099). Adjusted odds of anaemia were 3.24-fold higher in females (95% CI: 1.98–5.31, p < 0.0001). Widowed individuals had the highest prevalence (20.2%). Cotrimoxazole (CTX) prophylaxis reduced anaemia likelihood (14.3% vs. 18.3%, p = 0.005). Participants on INSTI-based ART had lower anaemia prevalence (11.9%) compared to protease

**Data availability statement:** All relevant data are within the paper and its Supporting Information files.

**Funding:** The author(s) received no specific funding for this work.

**Competing interests:** The authors have declared that no competing interests exist.

inhibitors (26.5%) or other regimens (30.0%). Viral load was independently associated with anaemia (AOR: 1.00, p = 0.021). Advanced WHO stages (Stage 3: 50% anaemia) and lower creatinine levels correlated with increased risk.

## Conclusions

Significant sex-based disparities in anaemia among Zambian PLHIV highlight the need for gender-responsive interventions, including nutritional support and INSTI-based ART. CTX prophylaxis demonstrates protective effects, advocating for broader integration into HIV care. Retrospective design and unmeasured confounders limit causal inference. Future research should prioritize longitudinal studies to refine anaemia management in high-burden settings.

## Introduction

Anaemia, defined by the World Health Organization (WHO) as haemoglobin levels below 13 g/dL in men and 12 g/dL in non-pregnant women, is a global public health issue affecting approximately 1.8 billion people worldwide [1]. Its prevalence varies significantly across regions and biological sex, with Sub-Saharan Africa (SSA) experiencing the highest burden 60% of children under five and 52% of pregnant women in SSA are anaemic, compared to global averages of 40% and 30%, respectively [2,3]. These disparities are driven by factors such as nutritional deficiencies (iron, folate, vitamin B12), infectious diseases (malaria, helminthiasis), and limited access to healthcare [3,4]. Among people living with HIV (PLHIV) on antiretroviral therapy (ART), anaemia is a frequent comorbidity, exacerbating health outcomes and mortality risks [5]. Globally, 30–40% of PLHIV on ART are affected by anaemia, with prevalence rates in SSA reaching 50–70%, according to a 2022 meta-analysis [6,7]. Sex-specific disparities are notable: studies in SSA report anaemia prevalence of 65–75% in women versus 45–55% in men on ART, reflecting biological, social, and structural factors [5,8]. For example, women face heightened risks due to menstrual blood loss, pregnancy-related iron demands, and sociocultural barriers to nutrient-rich diets. Men, however, often present with severe anaemia linked to advanced HIV progression and delayed ART initiation [9].

Epidemiological data highlight anaemia's clinical significance in HIV management [10]. A 2021 longitudinal study in SSA found that PLHIV with anaemia at ART initiation had a 50% higher risk of mortality within one year compared to non-anaemic patients, with sex-specific mortality trends: anaemic men exhibited a 60% increased risk of death versus 45% in women, likely due to later diagnosis and higher rates of opportunistic infections [11,12]. Furthermore, anaemia correlates with reduced ART adherence, slower immune recovery, and increased hospitalizations, disproportionately impacting women in resource-limited settings where caregiving roles and economic inequities limit healthcare access [13].

This study aims to comprehensively evaluate the prevalence of anaemia among people living with HIV (PLHIV) in Southern Province, Zambia, with a specific focus

on identifying sex-based disparities and elucidating the underlying clinical, nutritional, and sociodemographic factors associated with anaemia in this population. By stratifying data by sex and analyzing key determinants, the study seeks to generate evidence that can inform targeted interventions and improve clinical management strategies for anaemia among PLHIV in resource-limited settings.

## Methodology

### Study design

This was a retrospective cohort study of PLHIV enrolled in HIV care in 12 President's Emergency Plan for AIDS Relief (PEPFAR)-supported districts in one province of Zambia.

### Study setting

The study was conducted in southern province, Zambia. The research spanned 12 districts including: Chikankata, Choma, Kalomo, Kazungula, Livingstone, Mazabuka, Monze, Namwala, Pemba, Siavonga, Sinazongwe, and Zimba. The ART department records were utilized due to their systematic documentation of variables in smart care [14].

### Eligibility and recruitment criteria

From 6410 files available for abstraction, 3410 medical records were screened for PLHIV aged who were ≥15 years. A total of 2840 medical records met the inclusion criteria and were included in the analysis. 3000 records were excluded due to incomplete information, such as missing details on age, sex, marital status and 570 files had missing outcome, diagnosis or the outcome.

### Data collection

Clinical data for PLHIV were abstracted either from HIV paper-based or electronic medical records known as SmartCare in 45 health facilities located in 12 districts of Southern Province onto the Electronic Data Capture (REDCap) software. The patients outcome was recorded. In addition to clinical data ART regimen, WHO stage, BMI, duration on ART, retention, laboratory such as CD4 count and HIV viral load, and pharmacy-related data, were collected from laboratory and pharmacy registers, respectively, in facilities where SmartCare was not fully functional. Where possible, the data were triangulated from both SmartCare and paper registers [15]. A review of medical records was conducted from November 10, 2024, to December 19, 2024.

### Study variables

The outcome variable was anaemia in people living with HIV and was defined as women having haemoglobin (Hb) levels <12 g/dL, and <13 g/dL for men [16]. The independent variables included age, sex, facility, cohort, marital status, blood pressure (systolic and diastolic), viral load, CD4 count, WHO staging, ART regimen, height, and weight.

### Data analysis

We exported data from the REDCap application to Microsoft Excel for cleaning and thereafter analysed in statcrunch. Data were described using frequency and percentages for categorical variables and medians (interquartile ranges) for continuous variables. To test for normality, the Shapiro–Wilk test was used. The Wilcoxon rank-sum test was used to ascertain the statistical difference between the two medians. A relationship between two categorical variables was determined using a chi-squared test. Multivariable logistic regression was used to examine the factors associated with anaemia in PLHIV. The variables age, sex, facility location, marital status, WHO stage, ART regimen, and BMI in the adjusted analysis were selected based on previous literature and knowledge from HIV treatment and management experts. The

variables included in the final model were selected based on findings from previous studies [15]. To ascertain statistical significance, a p-value of <0.05 was used.

### Ethical considerations and consent to participants

Ethical approval for the data used in this study was obtained from the Mulungushi University School of Medicine and Health Sciences (SOHMS) Research Ethics Committee (ethics

  Reference number SMHS-MU2-2024-86) on 23rd July 2024. No information leading to identification of patients during and after analysis was abstracted and entered in the data collection form. Secondary data were used in this project. A written/verbal consent was not applicable and was therefore waived by the ethics committee.

  We used the Strengthening the Reporting of Observational Studies in Epidemiology (STROBE) checklist to guide reporting, Supplementary file S1 File.

## Results

From 6410 files available for abstraction, 3410 medical records were reviewed, 2840 records met the inclusion criteria and were included in the analysis. 3000 records were excluded due to incomplete information, such as missing details on age, sex, marital status and 570 files had missing outcome, diagnosis or the outcome, **Fig 1**.

### Relationship between demographic and clinical characteristic and Anaemia

Table 1 illustrates the relationship between anaemia with other study variables. The study included 2,840 participants, with 15.7% (n = 446) diagnosed with anaemia. Females constituted the majority (61.3%, n = 1,741) and exhibited a dispropor-tionately higher anaemia prevalence (21.8%, n = 380) compared to males (6.0%, n = 66). Marital status revealed nota-ble trends: widowed individuals had the highest anaemia prevalence (20.2%, n = 44/218), followed by divorced (16.6%,

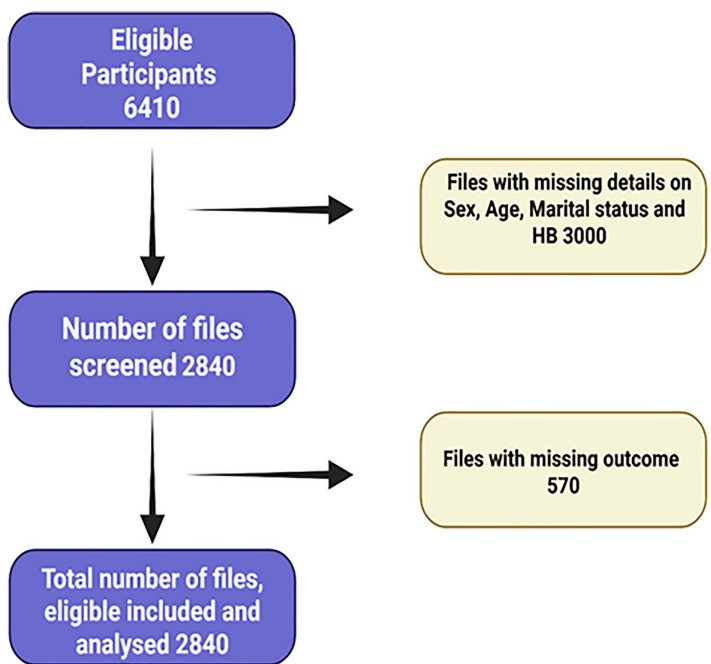

**Fig 1. Eligibility flow diagram.**

**Table 1. Relationship between demographic and clinical characteristics and Anaemia.**

| Variable | Median (IQR) or Frequency (%) | Anaemia | | P-Value |
|---|---|---|---|---|
| | | YES = 446 (15.7%) | No = 2394 (84.3) | |
| **Age, years** | 40 (33, 47) | 38 (32, 44) | 40 (33, 47) | **0.0003** |
| **Sex** | | | | |
| Male | 1,099 (38.7) | 66 (6.0) | 1,033 (94.0) | **<0.0001** |
| Female | 1,741 (61.3) | 380 (21.8) | 1,361 (78.2) | |
| **Marital status** | | | | |
| Never married | 374 (13.2) | 45 (12.0) | 329 (88.0) | 0.128 |
| Married | 1,691 (59.4) | 261 (15.4) | 1,430 (84.6) | |
| Divorced | 344 (12.1) | 57 (16.6) | 287 (83.4) | |
| Widowed | 218 (7.7) | 44 (20.2) | 174 (79.8) | |
| Unkown | 58 (2.0) | 10 (17.2) | 48 (82.8) | |
| **ART Regimen** | | | | |
| NNRTs/NRTISs | 2007 (70.7) | 336 (16.4) | 1,671 (83.3) | **0.001** |
| INSTIs | 764 (26.9) | 91 (11.9) | 673 (88.1) | |
| PIs | 49 (1.7) | 13 (26.5) | 36 (73.5) | |
| Other | 20 (0.7) | 6 (30.0) | 14 (70.0) | |
| **WHO clinical staging of HIV** | | | | |
| Stage 1 | 2758 (97.1) | 427 (15.5) | 2,331 (84.5) | **0.021** |
| Stage 2 | 18 (0.6) | 5 (27.8) | 13 (72.2) | |
| Stage 3 | 12 (0.4) | 6 (50) | 6 (50) | |
| Stage 4 | 2 (0.1) | 0 | 2 (100) | |
| **History of TB Before initiation of ART** | | | | |
| Yes | 37 (1.3) | 4 (10.8) | 33 (89.2) | 0.502 |
| No | 2,803 (98.7) | 442 (15.8) | 2,361 (84.2) | |
| **History of TB after initiation of ART** | | | | |
| Yes | 95 (3.4) | 82 (86.3) | 13 (13.7) | 0.582 |
| No | 2,745 (96.6) | 433 (15.8) | 2,312 (84.2) | |
| **History of CTX at initiation of ART** | | | | |
| Yes | 1,549 (54.5) | 249 (16.1) | 1,300 (83.9) | 0.552 |
| No | 1,291 (45.5) | 197 (15.3) | 1,094 (84.7) | |
| **Current on CTX at initiation of ART** | | | | |
| Yes | 190 (6.7) | 33 (17.4) | 157 (82.6) | 0.514 |
| No | 2,650 (93.3) | 413 (15.6) | 2,233 (84.4) | |
| **CTX as part of their treatment** | | | | |
| Yes | 1,823 (64.2) | 260 (14.3) | 1,563 (85.7) | **0.005** |
| No | 1,017 (35.8) | 186 (18.3) | 831 (81.7) | |
| **CTX before initation of treatment** | | | | |
| Yes | 655 (23.1) | 101 (15.4) | 554 (84.6) | 0.820 |
| No | 2,185 (76.9) | 345 (15.8) | 1,840 (84.2) | |
| **Treatment supporter spouse** | | | | |
| Yes | 1,326 (46.7) | 215(16.2) | 1,111 (83.8) | 0.485 |
| No | 1,514 (53.3) | 231 (15.3) | 1,283 (84.7) | |
| **Treatment supporter friend** | | | | |
| Yes | 23 (0.8) | 4 (17.4) | 19 (82.6) | 0.774 |
| No | 2,817 (99.2) | 442 (15.7) | 2,875 (84.3) | |

*(Continued)*

**Table 1.** (Continued)

| Variable | Median (IQR) or Frequency (%) | Anaemia | | P-Value |
|---|---|---|---|---|
| | | YES = 446 (15.7%) | No = 2394 (84.3) | |
| **Treatment supporter None** | | | | |
| *Yes* | 5 (0.2) | 0 | 5 (100) | 1 |
| *No* | 2,835 (99.8) | 446 (15.7) | 2,389 (84.3) | |
| **Treatment supporter other relatives** | | | | |
| *Yes* | 870 (30.6) | 139 (16.0) | 731 (84.0) | 0.791 |
| *No* | 1,970 (69.4) | 307 (15.6) | 1,663 (84.4) | |
| **Other** | | | | |
| *Yes* | 336 (11.8) | 41 (12.2) | 295 (87.8) | 0.061 |
| *No* | 2,504 (88.2) | 405 (16.2) | 2,099 (83.8) | |
| **Current SBP,** mmHg | 120 (110,132) | 118 (108, 128) | 120 (110, 132) | 1.000 |
| **Current DBP,** mmHg | 75 (67,83) | 73 (66, 82) | 75 (67, 83) | 1.000 |
| **Current BMI,** Body mass index, kg/m2 | 21.2 (18.8, 24) | 20.9 (17.9, 23.6) | 21.2 (18.9, 24.1) | 0.713 |
| **CD4,** cells/µL | 480 (316, 673) | 486 (303, 675) | 478.5 (318,673) | 0.992 |
| **Creatinine,**µmol/L | 70 (56, 84.8) | 65 (51.1, 81.7) | 71 (57, 85.4) | **<0.0001** |
| **Viral load,** copies/ml | 20 (0, 85) | 20 (0, 92) | 20 (0, 82) | 0.854 |
| **ALT,** µL | 25 (18, 34.9) | 21.3 (16, 31) | 25 (18.5, 35) | **<0.0001** |

**Abbreviations:** median (interquartile range), data presented as frequency (%), SBP; systolic blood pressure, DBP; diastolic blood pressure, BMI; body mass index, µL; micro perliter, µmol/L; micromole perliter, ART; Antiretroviral therapy, INSTIs; Integrase Strand Transfer Inhibitors.

NNRTIs; Non-Nucleoside Reverse Transcriptase Inhibitors, NRTIs; Nucleoside Reverse Transcriptase Inhibitors, p-value; Probability value.

n = 57/344) and married participants (15.4%, n = 261/1,691), while never-married individuals had the lowest prevalence (12.0%, n = 45/374). ART regimen differences were striking: most participants were on NNRTI/NRTI regimens (70.7%, n = 2,007), with 16.4% anaemic, while those on INSTIs (26.9%, n = 764) had lower anaemia prevalence (11.9%). In contrast, PI-based regimens (1.7%, n = 49) and "other" regimens (0.7%, n = 20) showed higher anaemia rates (26.5% and 30.0%, respectively). Advanced HIV progression (WHO Stage 2 and 3) correlated with elevated anaemia prevalence (27.8% and 50.0%), though sample sizes were small. Cotrimoxazole (CTX) use demonstrated a protective effect, with lower anaemia prevalence among those receiving CTX as part of treatment (14.3%, n = 260/1,823) compared to non-recipients (18.3%, n = 186/1,017). Treatment supporter type (spouse, relatives, or none) and TB history showed no significant differences in anaemia prevalence.

## Comparison between male and female demographic and characteristic of Anaemia

The median age of male participants was 43 years (IQR: 36–49), while for female participants it was 38 years (IQR: 31–45). Age was not significantly associated with anemia in either sex (p = 0.9329 in males; p = 0.171 in females). In both sexes, the majority of participants were married: 68.5% of males and 53.9% of females. Among married individuals, anemia prevalence was higher in females (23.1%) compared to males (5.8%). A higher proportion of widowed and divorced women were anemic (23.0% and 19.4%, respectively), compared to their male counterparts (5.7% and 11.2%). Most participants resided in urban areas 81.2% among males and 79.4% among females with no statistically significant difference in anemia prevalence based on residency for either sex. Regarding ART regimens, the majority of participants were on NNRTI-based therapy. A significantly higher proportion of males on NNRTIs were anemic compared to those on other regimens (7.3% vs. 3.4% on INSTIs; p = 0.003). Among females, anemia was also more common among those on PIs (42.9%) and other regimens (44.4%) than NNRTIs (21.3%) or INSTIs (21.9%), with a significant association (p = 0.041).

Most participants were classified as WHO Stage I at baseline, with 97.6% of males and 96.8% of females falling into this category. In women, a significant association was found between WHO stage and anemia (p = 0.027), with higher anemia prevalence in advanced stages. There was no significant association between history of TB (before or after ART initiation) and anemia in either sex. Likewise, cotrimoxazole (CTX) use whether before or at ART initiation, current use, or as part of treatment was not significantly associated with anemia in males. However, among females, those on CTX as part of treatment had a significantly lower anemia prevalence (19.6%) compared to those not on CTX (25.9%; p = 0.003). Among women, being supported by a spouse was associated with a higher prevalence of anemia (24.6%) compared to those without spousal support (19.9%, p = 0.020), while in men, there was no significant association between type of treatment supporter and anemia status. Regarding clinical measurements, female participants with anemia had a significantly lower median BMI (21.1 kg/m$^2$) compared to those without anemia (22.1 kg/m$^2$, p < 0.0001). Similarly, their median CD4 count was lower (495.5 vs. 538.5 cells/μL, p = 0.0174). ALT levels were also significantly lower in anemic females (20 U/L) compared to non-anemic females (23.1 U/L, p = 0.0002). No significant differences were found in blood pressure, creatinine, viral load, or ALT among male participants based on anemia status. **See Table 2**.

### Logistic regression of factors associated with anaemia in PLHIV

Table 3 shows the results of univariable and multivariable regression analyses of factors associated with anaemia in people in PLHIV. At univariable analysis, age (OR: 0.98, 95% CI: 0.97–0.99, p = 0.0006), sex (OR: 4.30, 95% CI: 3.27–5.64, p < 0.0001), marital status (OR: 1.10, 95% CI: 1.02–1.19, p = 0.0128), and viral load (OR: 1.00, 95% CI: 1.00–1.0000025, p = 0.878) were significantly associated with anaemia in people living with HIV (PLWHIV). In the multivariable model, sex (AOR: 3.24, 95% CI: 1.98–5.31, p < 0.0001), marital status (AOR: 1.19, 95% CI: 1.01–1.41, p = 0.0318), and viral load (AOR: 1.00, 95% CI: 1.00–1.00, p = 0.021) remained statistically significant after adjustment for other variables.

The prevalence of anaemia in this cohort of people living with HIV (PLHIV) was **15.7%**. This finding aligns with regional trends in Sub-Saharan Africa (SSA), where anaemia remains a persistent comorbidity despite advances in antiretroviral therapy (ART) [17]. This rate mirrors studies from Zambia, such as a 2020 Lusaka-based cohort reporting a 16% anaemia prevalence, and reflects broader patterns observed in Malawi (12–18%) and South Africa (14–22%) [8,18]. However, the multifactorial nature of anaemia in PLHIV driven by chronic inflammation, nutritional deficiencies, opportunistic infections, and ART-related hematotoxicity is particularly salient in Zambia, where overlapping burdens of malnutrition, endemic infections such as malaria, and socioeconomic inequities amplify risk [19].

After stratifying by sex we found that the prevalence of anaemia among PLWH was higher in females (21.8%, n = 380) than males (6.1%, n = 67). These findings are consistent with the study by Kamvuma et al., in Zambia which reported that females living with HIV were more likely to develop anaemia compared to their male counterparts [20]. Moreover, supporting our findings, a 2019 study conducted at the University Teaching Hospital in Lusaka, Zambia, reported significantly higher rates of anaemia among women living with HIV compared to men. This aligns with the sex-based disparities observed in our cohort, further highlighting that females living with HIV are at increased risk of developing anaemia [8,12,21]. Furthermore, a systematic review and meta-analysis by Azzam et al., examining the global, regional, and national prevalence of anaemia across 195 countries, consistently found higher anaemia rates among females [22]. This disparity is largely attributed to physiological factors such as menstruation, pregnancy, and childbirth. Similarly, studies conducted in sub-Saharan Africa, such as the one by Gebrerufael et al. in Ethiopia, have documented significantly higher anemia prevalence among women receiving ART [23]. This may be largely attributed to increased iron requirements, a higher incidence of chronic infections, and widespread nutritional deficiencies affecting women in the region. These challenges are exacerbated by systemic gender inequalities, including restricted access to nutritious diets and healthcare, often rooted in patriarchal norms that limit women's autonomy [24,25]. Similar this patterns is observed in Kenya and South Africa, where anaemia in women is worsened by overlapping health burdens such as HIV, malaria, and helminth infections [26].

**Table 2. Comparison between male and female demographic and characteristic of Anaemia.**

| Variable | Male | | | | Female | | | |
|---|---|---|---|---|---|---|---|---|
| | Median (IQR) or Frequency (%) | Yes = 67 (6.1%) | No = 1032 (93.9%) | P value | Median (IQR) or Frequency (%) | Yes = 380 (21.8%) | No = 1361 (78.2%) | P value |
| **Age, years** | 42 (36, 48) | 43 (36, 49) | 42 (36, 48) | 0.9329 | 38 (31, 45) | 38 (31, 44) | 38 (31, 46) | 0.171 |
| **Marital status** | | | | | | | | |
| _Never married_ | 124 (11.3) | 6 (4.8) | 118 (95.2) | 0.344 | 250 (14.4) | 39 (15.6) | 211 (84.4) | 0.080 |
| _Married_ | 753 (68.5) | 44 (5.8) | 709 (94.2) | | 938 (53.9) | 217 (23.1) | 721 (76.9) | |
| _Divorced_ | 107 (9.7) | 12 (11.2) | 95 (88.8) | | 237 (13.6) | 46 (19.4) | 191 (80.6) | |
| _Widowed_ | 35 (3.2) | 2 (5.7) | 33 (94.3) | | 183 (10.5) | 42 (23.0) | 141 (77.0) | |
| _Unknown_ | 18 (1.6) | 0 (0) | 18 (100) | | 40 (2.3) | 10 (25.0) | 30 (75.0) | |
| **Residency** | | | | | | | | |
| _Urban_ | 892 (81.2) | 52 (5.8) | 840 (94.2) | 0.443 | 1383 (79.4) | 300 (21.7) | 1083 (78.3) | 0.789 |
| _Rural_ | 207 (18.8) | 15 (7.3) | 192 (92.7) | | 358 (20.6) | 80 (22.4) | 278 (77.7) | |
| **ART Regimen** | | | | | | | | |
| _NNRTs/NRTIS_ | 648 (59.0) | 47 (7.3) | 601 (92.7) | **0.003** | 1359 (78.1) | 290 (21.3) | 1069 (78.7) | **0.041** |
| _INSTIs_ | 412 (37.4) | 14 (3.4) | 398 (96.6) | | 352 (20.2) | 77 (21.9) | 275 (78.1) | |
| _PIs_ | 28 (2.6) | 4 (14.3) | 24 (85.7) | | 21 (1.2) | 9 (42.9) | 12 (57.1) | |
| _Other_ | 11 (1.0) | 2 (18.2) | 9 (81.8) | | 9 (0.5) | 4 (44.4) | 5 (55.6) | |
| **WHO clinical staging of HIV** | | | | | | | | |
| _Stage 1_ | 1073 (97.6) | 63 (5.9) | 1010 (94.1) | 0.096 | 1685 (96.8) | 365 (21.7) | 1320 (78.3) | **0.027** |
| _Stage 2_ | 9 (0.8) | 2 (22.2) | 7 (77.8) | | 9 (0.5) | 3 (33.3) | 6 (66.7) | |
| _Stage 3_ | 5 (0.5) | 1 (20.0) | 4 (80.0) | | 7 (0.4) | 5 (71.4) | 2 (28.6) | |
| _Stage 4_ | 1 (0.1) | 0 (0) | 1 (100) | | 1 (0.1) | 0 (0) | 1 (100) | |
| **History of TB Before initiation of ART** | | | | | | | | |
| _Yes_ | 20 (1.8) | 0 (0) | 20 (100) | 0.628 | 17 (1.0) | 4 (23.5) | 13 (76.5) | 0.774 |
| _No_ | 1079 (98.2) | 67 (6.2) | 1012 (93.2) | | 1724 (99.0) | 376 (21.8) | 1348 (78.2) | |
| **History of TB after initiation of ART** | | | | | | | | |
| _Yes_ | 46 (4.2) | 3 (6.5) | 43 (93.5) | 0.756 | 49 (2.8) | 11 (22.5) | 38 (77.5) | 0.915 |
| _No_ | 1053 (95.8) | 64 (6.1) | 989 (93.9) | | 1692 (97.2) | 369 (21.8) | 1323 (78.2) | |
| **History of CTX at initiation of ART** | | | | | | | | |
| _Yes_ | 606 (55.1) | 41 (6.8) | 565 (93.2) | 0.304 | 943 (54.2) | 209 (22.2) | 734 (77.8) | 0.712 |
| _No_ | 493 (44.9) | 26 (5.3) | 467 (94.7) | | 798 (45.8) | 171 (21.4) | 627 (78.6) | |
| **Current on CTX at initiation of ART** | | | | | | | | |
| _Yes_ | 63 (5.7) | 5 (7.9) | 58 (92.1) | 0.582 | 127 (7.3) | 28 (22.1) | 99 (77.9) | 0.950 |
| _No_ | 1036 (94.3) | 62 (6.0) | 974 (94.0) | | 1614 (92.7) | 352 (21.8) | 1262 (78.2) | |
| **CTX as part of their treatment** | | | | | | | | |
| _Yes_ | 692 (63.0) | 38 (5.5) | 654 (94.5) | 0.274 | 1131 (65.0) | 222 (19.6) | 909 (80.4) | **0.003** |
| _No_ | 407 (37.0) | 29 (7.1) | 378 (92.9) | | 610 (35.0) | 158 (25.9) | 452 (74.1) | |
| **CTX before initiation of Treatment** | | | | | | | | |
| _Yes_ | 264 (24.0) | 15 (5.7) | 249 (94.3) | 0.747 | 391 (22.5) | 86 (22.0) | 305 (78.0) | 0.927 |
| _No_ | 835 (76.0) | 52 (6.2) | 783 (93.8) | | 1350 (77.5) | 294 (21.8) | 1056 (78.2) | |
| **Treatment supporter Spouse** | | | | | | | | |
| _Yes_ | 601 (54.7) | 37 (6.2) | 564 (93.8) | 0.927 | 725 (41.6) | 178 (24.6) | 547 (75.4) | **0.020** |
| _No_ | 498 (45.3) | 30 (6.0) | 468 (94.0) | | 1016 (58.4) | 202 (19.9) | 814 (80.1) | |

_(Continued)_

**Table 2.** (Continued)

| Variable | Male | | | | Female | | | |
|---|---|---|---|---|---|---|---|---|
| | Median (IQR) or Frequency (%) | Yes = 67 (6.1%) | No = 1032 (93.9%) | P value | Median (IQR) or Frequency (%) | Yes = 380 (21.8%) | No = 1361 (78.2%) | P value |
| **Treatment supporter friend** | | | | | | | | |
| Yes | 12 (1.1) | 0 (0) | 12 (100) | 1.000 | 11 (0.6) | 4 (36.4) | 7 (63.6) | 0.269 |
| No | 1087 (98.9) | 67 (6.2) | 1020 (93.8) | | 1730 (99.4) | 376 (21.7) | 1354 (78.3) | |
| **Treatment supporter None** | | | | | | | | |
| Yes | 2 (0.2) | 0 (0) | 2 (100) | 1.000 | 3 (0.2) | 0 (0) | 3 (100) | 1.000 |
| No | 1097 (99.8) | 67 (6.1) | 1030 (93.9) | | 1738 (99.8) | 380 (21.9) | 1358 (78.1) | |
| **Treatment supporter other relatives** | | | | | | | | |
| Yes | 253 (23.0) | 16 (6.3) | 237 (93.7) | 0.863 | 617 (35.4) | 124 (20.1) | 493 (79.9) | 0.196 |
| No | 846 (77.0) | 51 (6.0) | 795 (94.0) | | 1124 (64.6) | 256 (22.8) | 868 (77.2) | |
| **Other** | | | | | | | | |
| Yes | 114 (10.4) | 5 (4.4) | 109 (95.6) | 0.420 | 222 (12.8) | 36 (16.2) | 186 (83.8) | **0.030** |
| No | 985 (89.6) | 62 (6.3) | 923 (93.7) | | 1519 (87.2) | 344 (22.7) | 1175 (77.3) | |
| **Current SBP,** mmHg | 123 (113, 135) | 120 (111.5, 129.5) | 123 (113, 135) | 0.0568 | 119 (109, 129) | 118 (108, 128) | 119 (109, 130) | 0.1388 |
| **Current DBP,** mmHg | 76 (68, 84) | 73 (66.5, 83.5) | 76 (68, 84) | 0.3366 | 74 (66, 82) | 73 (65, 82) | 74 (67, 82) | 0.1715 |
| **Current BMI,** Body mass index, kg/m2 | 20.4 (18.5, 22.8) | 20.2 (17.6, 23.0) | 20.5 (18.5, 22.7) | 0.2344 | 21.8 (19.2, 25.1) | 21.1 (17.9, 23.7) | 22.1 (19.5, 25.4) | **<0.0001** |
| **CD4,** cells/µL | 407 (268, 584) | 408.5 (221, 596) | 407 (270, 578) | 0.5936 | 529.5 (357.5, 728) | 495.5 (324, 681.5) | 538.5 (366, 733) | **0.0174** |
| **Creatinine,** µmol/L | 78 (65, 95.7) | 82 (63.5, 107.9) | 78 (65, 95.4) | 0.4794 | 65 (42, 77.3) | 62.5 (50, 76.5) | 65 (52.1, 77.4) | 0.101 |
| **Viral load,** copies/ml | 20 (0, 108) | 33 (5, 130) | 20 (0, 103) | 0.1633 | 20 (0, 70) | 20 (0, 85) | 20 (0, 57) | 0.835 |
| **ALT,** µL | 28 (20.3, 39.3) | 26.4 (21.1, 34.9) | 28 (20, 40) | 0.8396 | 23 (17, 30.6) | 20 (15.5, 30) | 23.1 (17.8, 31.0) | **0.0002** |

Abbreviations: median (interquartile range), data presented as frequency (%), SBP; systolic blood pressure, DBP; diastolic blood pressure, BMI; body mass index, µL; micro perliter, µmol/L; micromole perliter, ART; Antiretroviral therapy, INSTIs; Integrase Strand Transfer Inhibitors.

NNRTIs; Non-Nucleoside Reverse Transcriptase Inhibitors, NRTIs; Nucleoside Reverse Transcriptase Inhibitors, p-value; Probability value.

**Table 3. Logistic regression of factors associated with anaemia in PLHIV.**

| Variable | Univariable analysis | | Multivariable analysis | |
|---|---|---|---|---|
| | OR (95%CI) | P. value | AOR (95%CI) | P. value |
| Age, years | 0.98 (0.97-0.99) | **0.0006** | 0.98 (0.96-1.01) | 0.155 |
| Sex | | | | |
| Males | REF | REF | REF | REF |
| Females | 4.300(3.27-5.64) | **<0.0001** | 3.24(1.98, 5.31) | **<0.0001** |
| Marital status | | | | |
| Unmarried | REF | REF | REF | REF |
| Married | 1.10 (1.02−1,19) | **0.0128** | 1.19 (1.01, 1.41) | **0.0318** |
| BMI | 0.99 (0.99,1.002) | 0.879 | 1.00(0.99, 1.00) | 0.532 |
| Viral load | 1.00 (1.00-1.0000025) | 0.878 | 1.00 (1.00, 1.00) | **0.021** |

Abbreviations: OR; odds ratio, AOR; adjusted odds ratio, BMI; body Mass Index, p-value; Probability value, REF; reference.

Marital status was another variable that was significant in our study. In studies conducted in Uganda and Tanzania, marital roles often correlate with heightened caregiving duties and financial stress, which can strain resources for nutrition and healthcare, thereby increasing anaemia risk [27,28]. However, a Nigerian study found that marriage can be protective, as spousal collaboration enhances access to food and medical services [29]. This contrast underscores the influence of cultural and socioeconomic contexts on marital dynamics [29,30]. In Zambia, where marriage may involve both shared resources and caregiving burdens, qualitative research is essential to unravel how these dynamics affect anaemia risk in HIV-affected households [31]. Such insights could guide community-based interventions targeting economic disparities and psychosocial stressors [32].

Lastly, viral load was another variable that was worth noting. The role of viral load in anaemia is supported by biological evidence across SSA [33]. In Malawi, elevated HIV viral loads increase pro-inflammatory cytokines (e.g., IL-6, TNF-α), disrupting red blood cell production and iron regulation [34]. A Zambian clinical trial demonstrated that early initiation of antiretroviral therapy (ART) reduces anaemia incidence by suppressing viral replication and inflammation [20]. However, the modest impact of viral load alone suggests multifactorial causes, including opportunistic infections such as tuberculosis and micronutrient deficiencies, as noted in Zimbabwean studies [35–37]. This highlights the need for integrated approaches combining viral suppression, nutritional support, and infection control to address anaemia comprehensively [38].

### Strengths

The study's large sample size of 2,840 participants across 12 districts in Southern Zambia enhances its statistical power and regional representativeness, providing a robust foundation for understanding anaemia trends among people living with HIV (PLHIV). The inclusion of sociodemographic, clinical, and laboratory variables enabled a comprehensive analysis of anaemia determinants, while multivariable logistic regression strengthened causal inference by adjusting for confounders. Notably, the strong association between female sex and anaemia (AOR: 3.24, 95% CI: 1.98–5.31) persisted after adjustment, aligning with regional evidence from Zambia, Malawi, and South Africa. This consistency underscores the study's relevance to Sub-Saharan Africa (SSA), where gender inequities and overlapping health burdens amplify anaemia risk. The identification of cotrimoxazole's (CTX) protective effect (14.3% vs. 18.3% anaemia prevalence) offers actionable insights for scaling prophylaxis programs in resource-limited settings.

### Limitations

The study's retrospective design and reliance on medical records introduced risks of selection bias, particularly due to the exclusion of 28% of screened data (1,073 records) for missing haemoglobin values. This exclusion may have skewed results if omitted individuals differed systematically (e.g., higher anaemia prevalence in excluded groups). Key limitations include imprecise estimates for variables with wide confidence intervals (CIs), such as marital status (AOR: 1.19, 95% CI: 1.01–1.41), which reflect heterogeneity in marital dynamics (e.g., caregiving burdens vs. spousal support) not captured by the cross-sectional design. Similarly, the negligible effect size of viral load (AOR: 1.00, 95% CI: 1.00–1.00) despite statistical significance suggests limited clinical relevance, likely due to residual confounding or measurement limitations. Small subgroup sizes further reduced precision: advanced HIV (WHO Stage 3, n = 12) and protease inhibitor users (n = 49) had unstable estimates (e.g., 50% anaemia prevalence in Stage 3), limiting reliable conclusions. Unmeasured confounders, such as nutritional status and malaria co-infection, likely contributed to residual bias, while the lack of etiological biomarkers such as ferritin obscured anaemias' heterogeneous causes.

### Conclusion

This study identifies a 15.7% prevalence of anaemia among people living with HIV (PLHIV) in Southern Zambia, with significant sex disparities females faced 3.24 fold higher adjusted odds of anaemia than males, driven by biological

vulnerabilities (menstrual blood loss) and systemic gender inequities (limited healthcare access). The multifactorial aetiology of anaemia, influenced by marital dynamics, advanced HIV progression, and viral load, underscores the need for integrated interventions, including gender-responsive nutrition programs, optimized ART regimens (e.g., INSTIs), and expanded cotrimoxazole prophylaxis. While the study's large, multisite cohort enhances regional relevance, retrospective design and unmeasured confounders (e.g., nutritional biomarkers) limit causal inference. Future research should prioritize longitudinal, biomarker-integrated studies to refine anaemia management in high-burden Sub-Saharan African settings, advancing equitable HIV care and global health targets.

## Supporting information

**S1 File. Strobe checklist.**
(DOCX)

**S2 File. Raw Data.**
(XLSX)

## Author contributions

**Conceptualization:** Martin Chakulya, Sepiso K. Masenga.

**Data curation:** Martin Chakulya, Sepiso K. Masenga.

**Formal analysis:** Martin Chakulya, Lweendo Muchaili, Hanzooma Hatwiko, Geofrey Mupeta, Joreen P. Povia, Sepiso K. Masenga.

**Investigation:** Martin Chakulya, Hanzooma Hatwiko, Matenge Mutalange, Bislom C. Mweene, Geofrey Mupeta, Sepiso K. Masenga.

**Methodology:** Martin Chakulya, David Chisompola, Lukundo Siame, Sepiso K. Masenga.

**Project administration:** Martin Chakulya, Benson M. Hamooya, Sepiso K. Masenga.

**Resources:** Martin Chakulya, Lweendo Muchaili, Matenge Mutalange, Memory Ngosa, Bislom C. Mweene, David Chisompola, Lukundo Siame, Joreen P. Povia, Benson M. Hamooya, Sepiso K. Masenga.

**Software:** Martin Chakulya, Memory Ngosa, Joreen P. Povia, Benson M. Hamooya, Sepiso K. Masenga.

**Supervision:** Martin Chakulya, Bislom C. Mweene, Geofrey Mupeta, David Chisompola, Marshall C. Mubanga, Lukundo Siame, Joreen P. Povia, Benson M. Hamooya, Sepiso K. Masenga.

**Validation:** Martin Chakulya, Hanzooma Hatwiko, Memory Ngosa, Geofrey Mupeta, Marshall C. Mubanga, Lukundo Siame, Sepiso K. Masenga.

**Visualization:** Martin Chakulya, Hanzooma Hatwiko, Matenge Mutalange, Memory Ngosa, Geofrey Mupeta, David Chisompola, Marshall C. Mubanga, Joreen P. Povia, Sepiso K. Masenga.

**Writing – original draft:** Martin Chakulya, Lweendo Muchaili, Hanzooma Hatwiko, Matenge Mutalange, Bislom C. Mweene, Marshall C. Mubanga, Lukundo Siame, Joreen P. Povia, Benson M. Hamooya, Sepiso K. Masenga.

**Writing – review & editing:** Martin Chakulya, Lweendo Muchaili, Lukundo Siame, Benson M. Hamooya, Sepiso K. Masenga.

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
