## [Decision Letter · Decision Letter 0]

23 Jun 2025

PONE-D-25-24917Anaemia and associated factors in people living with HIV on ART in Southern Province of ZambiaPLOS ONE

Dear Dr. CHAKULYA,

Thank you for submitting your manuscript to PLOS ONE. After careful consideration, we feel that it has merit but does not fully meet PLOS ONE’s publication criteria as it currently stands. Therefore, we invite you to submit a revised version of the manuscript that addresses the points raised during the review process.

**ACADEMIC EDITOR: **

This research is quite relevant to the research community, however, authors are advised to incorporate and respond to the comments by both reviewers to strengthen their paper. Particular focus should be on the suggestions to improve the results and discussion sections.  

We look forward to receiving your revised manuscript.

Kind regards,

Ewurama Dedea Ampadu Owusu, PhD

Academic Editor

PLOS ONE

Journal Requirements:

Reviewers' comments:

Reviewer's Responses to Questions

**Comments to the Author**

1. Is the manuscript technically sound, and do the data support the conclusions?

Reviewer #1: Yes

Reviewer #2: Yes

2. Has the statistical analysis been performed appropriately and rigorously? 

Reviewer #1: Yes

Reviewer #2: I Don't Know

3. Have the authors made all data underlying the findings in their manuscript fully available?

Reviewer #1: Yes

Reviewer #2: Yes

4. Is the manuscript presented in an intelligible fashion and written in standard English?

Reviewer #1: Yes

Reviewer #2: Yes

5. Review Comments to the Author

Reviewer #1: The manuscript presents a rigorous and well-supported scientific investigation, with data that substantiates its conclusions. The introduction is comprehensive, offering a detailed background, while the study design is appropriately structured to address the research objectives. The methodology is sufficiently detailed to ensure reproducibility, and the data analysis is both suitable and meticulously performed. To enhance clarity, the authors should incorporate a separate table exclusively presenting demographic characteristics, stratified by gender and including P values, positioned prior to Table 2. Additionally, a subheading “Results” should be inserted at line 179 to distinctly delineate the discussion section. The numbering of tables requires verification for accuracy and consistency, and all supporting data is adequately provided. The manuscript is articulated in precise, unambiguous, and grammatically sound English, ensuring the effective dissemination of scientific findings. The conclusions drawn in the study are directly based on the data presented, reinforcing the validity and reliability of the research outcomes. Furthermore, the research adheres to all applicable standards for ethical experimentation and research integrity, ensuring compliance with established guidelines for responsible scientific conduct.

Reviewer #2: The manuscript is technically sound and the data supports the conclusions that were made. The statistical analysis seem to be in order. With respect to data availability, the authors have indicated that some restrictions will apply. However, they have uploaded the raw data underlying the results presented in the study as supporting information S2. The manuscript is largely presented in an intelligible fashion and written in standard English. A few corrections that must be made have been detailed in the attached review.

6. PLOS authors have the option to publish the peer review history of their article (what does this mean? ). If published, this will include your full peer review and any attached files.

**Do you want your identity to be public for this peer review?** For information about this choice, including consent withdrawal, please see our Privacy Policy .

Reviewer #1: No

Reviewer #2: No

---

## [Author Response · Author response to Decision Letter 1]

11 Jul 2025

10/07/2025

PLOS ONE Journal

Dear Editor,

Ref: Submission of a revised research article for peer review and publication consideration

Reference to the above-mentioned subject. I am writing to submit a revised original research article titled "Anaemia and associated factors in people living with HIV on ART in Southern Province of Zambia,".

We would like to thank the reviewers for taking the time to make suggestions that have improved our manuscript. We have revised the manuscript and addressed all concerns and suggestions. We now hope the current manuscript is acceptable for publication. Below are the point-by-point responses to all comments and suggestions.

Response to the reviewer: No. 1

The objectives outlined in the introduction on page 4 lines 78-80 (“This study evaluates how combined strategies routine haemoglobin checks, gender-responsive nutrition support, and infection-specific treatment can reduce anaemia-related illness and deaths among people with HIV on antiretroviral therapy in Sub-Saharan Africa”) should be rewritten to align with those in the abstract (Page 2 line 27-29). It is a better reflection of what the study set out to do.

Response: Thank you very much for the suggestion: we have revised the manuscript the manuscript as suggested.

Reviewer comments: No. 2

Rephrase lines 81-83 (page 4). The study findings clearly highlight gender-based disparities which must be addressed in order to improve HIV care and lessen the burden of anaemia. However, addressing these disparities per se were not within the scope of the study. In view of this the remarks on page 4 lines 81 and 83 “By addressing gender-based disparities and systemic health inequities, it [i.e. the study] aims to create scalable solutions to improve HIV care, advance global HIV goals, and lessen the dual burden of anaemia and inequality in high-prevalence regions” should be rephrased.

Response: We have revised the manuscript the manuscript as suggested. We included a table (table 2) showing gender based disparities.

Reviewer comments: No. 3

Page 6 line 149- reference should be made to Table 1 and not Table 2.

Response: Thank you very much for the suggestion: we have revised the manuscript the manuscript as suggested.

Reviewer comments: No. 4

Page 9 line 177-178. The title for Table 2 includes the phrase, ‘…haemoglobin concentration at 12 months’ however it is not immediately clear what the 12 months is/signifies. A brief explanation can be included as a footnote to the table.

Response: Thank you very much for the suggestion: we have revised the manuscript the manuscript as suggested. We removed the contradicting statement which was 12 months

Reviewer comments: No. 5

Page 10 line 190. Please check reference 20. It is unclear what information was cited from it. The citation may be erroneous.

Response: Thank you very much for the suggestion: we have revised the manuscript the manuscript as suggested. The reference was worked on.

We have revised the manuscript and addressed all concerns raised. We want to thank you all again for the tremendous work and time that you committed to reviewing and correcting our work. Our manuscript is much improved, and we are very grateful.

Please address all correspondence to chakulyamartin1@gmail.com . We look forward to hearing from you at your earliest convenience.

Please do not hesitate to contact me should you have further questions.

Yours sincerely,

Mr Chakulya Martin, BSc., MSc.

Lecturer Mulungushi University Teaching Hospital

---

## [Decision Letter · Decision Letter 1]

20 Aug 2025

Anaemia and associated factors in people living with HIV on ART in Southern Province of Zambia

PONE-D-25-24917R1

Dear Dr. CHAKULYA,

We’re pleased to inform you that your manuscript has been judged scientifically suitable for publication and will be formally accepted for publication once it meets all outstanding technical requirements.

Kind regards,

Ewurama Dedea Ampadu Owusu, PhD

Academic Editor

PLOS ONE

Additional Editor Comments (optional):

Reviewers' comments:

Reviewer's Responses to Questions

**Comments to the Author**

1. If the authors have adequately addressed your comments raised in a previous round of review and you feel that this manuscript is now acceptable for publication, you may indicate that here to bypass the “Comments to the Author” section, enter your conflict of interest statement in the “Confidential to Editor” section, and submit your "Accept" recommendation.

Reviewer #1: All comments have been addressed

Reviewer #2: All comments have been addressed

2. Is the manuscript technically sound, and do the data support the conclusions?

Reviewer #1: Yes

Reviewer #2: Yes

3. Has the statistical analysis been performed appropriately and rigorously? 

Reviewer #1: Yes

Reviewer #2: I Don't Know

4. Have the authors made all data underlying the findings in their manuscript fully available?

Reviewer #1: Yes

Reviewer #2: Yes

5. Is the manuscript presented in an intelligible fashion and written in standard English?

Reviewer #1: Yes

Reviewer #2: Yes

6. Review Comments to the Author

Reviewer #1: All concerns raised in the initial review have been thoroughly addressed in the revised manuscript. The study is original, methodologically sound, and aligns with the scientific and editorial standards required for publication in the journal.

Reviewer #2: (No Response)

7. PLOS authors have the option to publish the peer review history of their article (what does this mean? ). If published, this will include your full peer review and any attached files.

**Do you want your identity to be public for this peer review?** For information about this choice, including consent withdrawal, please see our Privacy Policy .

Reviewer #1: No

Reviewer #2: No

---

## [Editor Report · Acceptance letter]

PONE-D-25-24917R1

PLOS ONE

Dear Dr. Chakulya,

I'm pleased to inform you that your manuscript has been deemed suitable for publication in PLOS ONE. Congratulations! Your manuscript is now being handed over to our production team.

Kind regards,

on behalf of

Dr. Ewurama Dedea Ampadu Owusu

Academic Editor

PLOS ONE